Evaluation of SARS-CoV-2 identification methods through surveillance of companion animals in SARS-CoV-2-positive homes in North Carolina, March to December 2020

Gin Taylor E. tegin@ncsu.edu 1
Petzold Elizabeth A. 2
Uthappa Diya M. 3
Neighbors Coralei E. 3
Borough Anna R. 3
Gin Craig 1
Lashnits Erin 4
Sempowski Gregory D. 5
Denny Thomas 5
Bienzle Dorothee 6
Weese J. Scott 6
Callahan Benjamin J. 1
Woods Christopher W. 2 3 5
1 College of Veterinary Medicine, North Carolina State University , Raleigh , NC , United States of America
2 Department of Infectious Diseases, Duke University , Durham , NC , United States of America
3 Hubert-Yeargan Center for Global Health, Duke University , Durham , NC , United States of America
4 Department of Medical Sciences, School of Veterinary Medicine, University of Wisconsin-Madison , Madison , WI , United States of America
5 Duke Human Vaccine Institute, Duke University , Durham , NC , United States of America
6 Ontario Veterinary College, University of Guelph , Guelph , Ontario , Canada
Kistler Whitney
Electronic publication date: 2023 Oct 24
Publication date: 2023
Volume: 11
Electronic Location ID: e16310
Received 2023 May 8; Accepted 2023 Sep 27
Copyright: ©2023 Gin et al.
Copyright year: 2023
Copyright holder: Gin et al.
License: This is an open access article distributed under the terms of the Creative Commons Attribution License, which permits unrestricted use, distribution, reproduction and adaptation in any medium and for any purpose provided that it is properly attributed. For attribution, the original author(s), title, publication source (PeerJ) and either DOI or URL of the article must be cited.
License URL: https://creativecommons.org/licenses/by/4.0/

Keywords: COVID-19, Coronavirus, SARS-CoV-2, Dogs, Cats, Antibody, Serology

Funding: An NIEHS-funded predoctoral fellowship to Taylor Gin (T32 OD011130) The NC DHHS (North Carolina Session Law 2020-4, An Act to Provide Aid to North Carolinians in Response to the Coronavirus Disease 2019 Crisis) COVID-19 samples were processed under BSL2 with aerosol management enhancement or BSL3 in the Duke Regional Biocontainment Laboratory which received partial support for construction from NIH/NIAID UC6AI058607 This publication was made possible by an NIEHS-funded predoctoral fellowship to Taylor Gin (T32 OD011130), and funds from the NC DHHS (North Carolina Session Law 2020-4, An Act to Provide Aid to North Carolinians in Response to the Coronavirus Disease 2019 Crisis). COVID-19 samples were processed under BSL2 with aerosol management enhancement or BSL3 in the Duke Regional Biocontainment Laboratory which received partial support for construction from NIH/NIAID (UC6AI058607). The contents are solely the responsibility of the authors and do not necessarily represent the official views of the NIEHS, or NC DHHS. The funders had no role in study design, data collection and analysis, decision to publish, or preparation of the manuscript.

==============================
We collected oral and/or rectal swabs and serum from dogs and cats living in homes with SARS-CoV-2-PCR-positive persons for SARS-CoV-2 PCR and serology testing. Pre-COVID-19 serum samples from dogs and cats were used as negative controls, and samples were tested in duplicate at different timepoints. Raw ELISA results scrutinized relative to known negative samples suggested that cut-offs for IgG seropositivity may require adjustment relative to previously proposed values, while proposed cut-offs for IgM require more extensive validation. A small number of pet dogs (2/43, 4.7%) and one cat (1/21, 4.8%) were positive for SARS-CoV-2 RNA, and 28.6 and 37.5% of cats and dogs were positive for anti-SARS-CoV-2 IgG, respectively.

Introduction

Novel severe acute respiratory syndrome coronavirus 2 (SARS-CoV-2) resulted in the coronavirus disease 2019 (COVID-19) pandemic, which has an ongoing, significant impact on human health and wellbeing. Regionally, at the time and place where this study was conducted, cases rose drastically from 101 reported cases per week in March 2020 to 40,407 cases per week in December 2020 (North Carolina Department of Health and Human Services (DHHS), 2020). Since the first documented companion animal case of SARS-CoV-2 in March 2020, many reports of SARS-CoV-2 RNA and/or antibody detection in pet dogs and cats have been reported (Newman et al., 2020; Barua et al., 2021; Dileepan et al., 2021; Hamer et al., 2021; Murphy & Ly, 2021; Fritz et al., 2021; Cossaboom et al., 2021; Goryoka et al., 2021; Bienzle et al., 2022). Transmission of SARS-CoV-2 to and/or from companion animals is particularly important since dogs and cats frequently share close proximity to their owners and often interact with people and other animals outside their household.

Previous studies have reported that companion animals develop an immune response to SARS-CoV-2 (Newman et al., 2020; Barua et al., 2021; Dileepan et al., 2021; Hamer et al., 2021; Murphy & Ly, 2021; Fritz et al., 2021; Cossaboom et al., 2021; Goryoka et al., 2021; Bienzle et al., 2022). However, due to the novel nature of COVID-19 and its unknown role in companion animals, serologic assays for animals had not been well validated at the time that many of these studies took place, and reported seropositivity ranges are highly variable (Newman et al., 2020; Barua et al., 2021; Dileepan et al., 2021; Hamer et al., 2021; Murphy & Ly, 2021; Fritz et al., 2021; Cossaboom et al., 2021; Goryoka et al., 2021; Bienzle et al., 2022). The varying sampling timelines, testing methods, and lack of well-defined controls within such studies to validate SARS-CoV-2 serology testing in animals leaves clearly defined seroprevalence in exposed pet populations unresolved.

Here we applied new methods for interpreting SARS-CoV-2 serology results in pet dogs and cats from SARS-CoV-2-PCR-positive homes through replicate measurements in target populations and pre-COVID-19 control samples. With these outcomes, we assessed the relationship between owner-answered questionnaires on pet behavioral and demographic data with seropositivity. Additionally, we measured SARS-CoV-2 RNA in samples using established methodology.

Materials and Methods

Ethics statement

The authors confirm that the ethical policies of the journal, as noted on the journal’s author guidelines page, have been adhered to and the appropriate ethical review committee approval has been received.

Study design and setting

This was a prospective, observational descriptive study. Households were recruited from an approximately 100-mile radius surrounding Durham, NC, from March 2020–March 2021. A written consent form was obtained for each study participant and pet enrolled in the study.

Sample collection

This study was a collaboration between Duke University and North Carolina State University, and was part of an ongoing study known as the Molecular and Epidemiological Surveillance in Suspected Infection (MESSI) study. Human sample collection was approved by the Duke Institutional Review Board (IRB) under the MESSI protocol number Pro00100241. Animal collection protocols were reviewed and approved by the Duke Institutional Animal Care and Use Committee (IACUC) before approaching an owner for permission to collect samples under the Molecular and Epidemiological Surveillance in Suspected Infection in House Pets (MESSI-HP) protocol number A079-20-03. The MESSI study team consisted of lab members with expertise in epidemiological field research and human sample collection. The MESSI-HP study team included one member of the MESSI field team with animal handling experience, one veterinary technician, and one veterinarian, who was present for house pet serum sample and oral and/or rectal swab collection. Inclusion and exclusion criteria are listed in Table 1 for humans and house pets.

Table 1 Inclusion and exclusion criteria for humans, dogs, and cats to participate in surveillance of SARS-CoV-2 through the MESSI and MESSI-HP studies.

Subject	Inclusion criteria	Exclusion criteria	
Humans	(a) Aged 2 years or older
(b) Weight > 6.4 kg (14lb)
(c) Subjects who meet at least one of the following populations in the community, or who present to the emergency room, outpatient clinics, or who are hospitalized:
• Subjects with current or historical symptoms of suspected infection, or symptoms that mimic infectious illness
• Subjects with exposure to someone with symptoms of suspected infection
• Subjects with confirmed infection (with or without symptoms)
• Subjects with recent vaccination/planned vaccination
(d) Ability of the subject or legally authorized representative/parent to understand study procedures, and willing and able to comply with all required procedures	Subjects will be excluded from the study if they meet ANY of the following criteria:
(a) Any specific condition that in the judgment of the referring provider or the site investigator precludes participation because it could affect subject safety or ability of subject to participate in this trial	
House pets	(a) At least one cohabitating household person enrolled in the MESSI study
(b) Owner willingness for pet to participate
(c) Ability of the subject or legally authorized representative/parent to understand study procedures, and willing and able to comply with all required procedures
(d) Pet safely able to be handled by study team	(a) As above	
Notes.

Abbreviations kg kilograms

lb pounds

The MESSI study team collected serial biological samples (nasopharyngeal swabs, serum samples) of humans throughout the course of their illness (days 0 (day of enrollment in MESSI study), 1, 3, 7, 14, 21, 28, 45, 60, 120).

Biological samples (oral swabs, rectal swabs, serum samples) were collected from house pets (dogs and cats) on visit days 0, 1, 3, 7, 14, 21, 28, 45, 60, 120, and/or 180. Some sample collection timepoints were omitted due to challenges with pet compliance or insufficient team members present who were comfortable assisting in animal sample collection. Oral and/or rectal swabs from exposed house pets were collected from days 0–60. Serum samples from exposed house pets were collected on day 28 or later to allow for seroconversion and to limit the number of people exposed to an actively infected or clinically ill human.

Oral swabs were collected by placing a sterile polyester-tipped applicator (manufacturer Puritan 25-806-1P) into the house pet’s mouth and rubbing the cheeks, gums, and tongue. Rectal swabs were collected by placing an applicator into the rectum and swabbing gently in a circular motion. Polyester-tipped applicators were immediately placed in viral transport media (VSM01) from Dasky (components included Hanks’ balanced salts, sucrose, penicillin, gentamicin, streptomycin sulfate, amphotericin B, nonessential amino acid, and phenol red) in a disposable tube for transport and storage. Serum samples were collected using standard venipuncture and processing for transport and storage.

Sample types and labels

Exposed serum samples

Serum samples collected from house pets of SARS-CoV-2-positive households are referenced as “exposed serum samples” throughout.

Pre-2019 serological controls

Forty-three (24 canine and 19 feline) pre-2019 serum samples submitted to the North Carolina State University’s Vector-Borne Disease and Diagnostic Laboratory (NCSU-VBDDL) for vector-borne disease testing prior to the emergence of SARS-CoV-2 (date of last sample April 19, 2018) were used as single-masked SARS-CoV-2-negative serological controls. Serum samples were frozen and stored at −20 °C until use in this study. Throughout the paper, these are referenced as “pre-2019 serological controls”.

Exposed PCR samples

Oral and rectal swab samples collected from house pets from SARS-CoV-2-positive households are referenced as “exposed PCR samples” throughout the paper.

Unexposed PCR controls

Ten oral/rectal swab samples from six house pets (three dogs and three cats) from SARS-CoV-2-PCR-negative homes were included in PCR testing. These samples came from house pets belonging to owners who were enrolled in the MESSI study based on the aforementioned inclusion/exclusion criteria but who were found to be negative for SARS-CoV-2 by PCR. Throughout the paper, these are referenced as “unexposed PCR controls”.

Polymerase chain reaction for SARS-CoV-2

To assess for the presence of viral SARS-CoV-2 RNA in exposed PCR samples and unexposed PCR controls, standard methodology was used as per the recommendation of the World Health Organization at the time (testing performed November 2021) (Diagnostic testing for SARS-CoV-2). Automated QIAsymphony (Qiagen LLC, Germantown, MD) two-step RT-PCR and World Health Organization E_Sabeco primer-probe sets (Charite, Berlin) were used (Corman et al., 2020). To evaluate the number of viral copies detected within a sample, quantitative PCR was carried out on a QuantStudio 3 Real-Time PCR System (Thermo Fisher Scientific, Waltham, MA). Samples with LOQ ≥ 62 RNA copies/mL (1.79 log10) from 800 µL of sample were considered positive based on a threshold determined by the manufacturer to indicate a positive test. To improve the accuracy in reporting positive tests, samples with LOQ ≥ 62 RNA copies/mL were subsequently run on the COBAS 6800 system, a qPCR test approved for emergency use by the FDA in 2021 to detect SARS-CoV-2 RNA in swab samples (Roche, Basel, Switzerland) (Roche Diagnostics, 2021). This test is typically reported qualitatively to users. The COBAS 6800 is a dual target assay where target 1 is a sequence-specific to SARS-CoV-2 and target 2 is more general to the Sarbeco virus subgenus. Detection of target 1 is considered indicative of the presence of SARS-CoV-2, whereas detection of target 2 typically indicates the presence of SARS-CoV-2 in low concentrations. Samples were considered positive for SARS-CoV-2 with a cycle threshold (Ct) value < 38 for target 1 or target 2 (Pujadas et al., 2020). Samples were only reported as positive if they were positive on both the QuantStudio 3 RT-PCR and COBAS-6800 systems. Additional PCR methods, including a description of primers, for the QuantStudio 3 RT PCR system are provided in supplementary materials (Supplemental Information 1).

Serology for SARS-CoV-2

Enzyme-linked immunosorbent assay (ELISA) was used for the detection of house pet IgG and IgM to SARS-CoV-2 in the pre-2019 serological controls and exposed serum samples as previously described by (Bienzle et al., 2022). Adsorption immunoassay plates (96-well, ThermoFisher, Mississauga, ON) were coated at 4 °C with 2 µg/mL of His-tagged SARS-CoV-2 S1 (GenScript, Piscataway, NJ) and incubated overnight. The following day, wells were washed 3x, blocked with 3% skim milk in Tris buffer for 60 min, washed 3x, and then 60 µL of five 3-fold dilutions (1:100, 1:300, 1:900, 1:2,700 and 1:8,100) of each serum sample was added. Plates were incubated for 120 min, washed 3x, and secondary antibodies conjugated to horseradish peroxidase (HRP) and diluted 1:5,000 were added for 60 min. Wells were washed 3x, and HRP activity was confirmed visually by adding trimethyl benzidine (TMB) substrate. Reactions were ceased with sulfuric acid, and optical density (O.D.) at 450 nm was read. Secondary antibodies were derived from goats and consisted ofanti-dog IgG, anti-dog IgM, anti-cat IgG, and anti-cat IgM (all from Abcam, Waltham, MA). Control samples for ELISA validation differed from the pre-2019 controls. For cats, the positive control ELISA validation sample consisted of serum from an experimentally infected SARS-CoV-2 cat (kindly provided by Y. Kawaoka, Madison, WI; positive feline control, used at 1:5,000 in ELISA), and negative controls included three different batches of pooled cat serum from 2016 or 2017, two serum samples from cats with feline infectious peritonitis (due to mutated feline enteric coronavirus), and one serum sample from a cat with osteomyelitis and hyperglobulinemia. For dogs, negative controls included three different batches of pooled dog serum collected in 2017, 2018, and 2019. Since serum from experimentally infected dogs was not available, the positive control for ELISA validation consisted of a serum sample from one study dog (exposed serum sample) with a high O.D. Each ELISA plate included 16 wells that were not coated with recombinant protein (blank), five replicate 1:100 dilutions of species-specific negative control samples, and five replicates of each of 3 dilutions of the positive control and test samples (1:100, 1:200, 1:400).

From the MESSI-HP serum samples and the pre-2019 serological controls, serum with a volume of at least 0.5 mL was divided into two aliquots, and the ELISA was performed on each aliquot. Staff performing the ELISA were masked to the identity of samples and group (pre-2019 serological control or exposed serum sample) for the second round of serology. These samples were used to calculate a false positive rate from initial reporting of “positive” and “negative”.

Questionnaires

Three questionnaires were provided to the primary owner. Each questionnaire was in paper copy and filled out either by a primary owner (an owner who assumed primary responsibility for general caretaking of the pet and lived with the pet full time), or by one of the study team members while verbally asking each question to the owner. When possible, the same primary owner provided answers at each visit. Questionnaire 1 (Supplemental Information 2 collected demographic information about each pet and was given to owners at the enrollment visit. Questionnaire 2 (Supplemental Information 3) collected clinical signs for each pet at every visit and contained 13 specific questions regarding the presence or absence of specific symptoms. Only questions with at least one “yes” and one “no” for each column were included for analysis. An additional column was added for the “presence” or “absence” of any symptom as dictated by the responses to the other symptom questions. Questionnaire 3 (Supplemental Information 4) contained 17 specific questions regarding human-animal interactions and animal behavior and was provided at the time of exposed serum sample collection, which occurred between days 28-180. Only questions free from errors in owner reporting that might have led to a spurious outcome were analyzed. Errors included owners circling multiple answers (as opposed to one answer) for multiple choice questions, failing to answer any part of the question, and/or answering the main question but failing to fill out related subquestions.

Statistical analysis

Statistical tests were conducted using R 4.1.1 (R Core Team, 2021). Descriptive statistics were calculated for the primary outcome of the percent of exposed dogs and cats positive on PCR and serology, as well as median and range of animal age. Lin’s concordance correlation coefficient for agreement on continuous measures was used to determine agreement between round 1 and round 2 O.D.s. Fisher’s exact test was used for univariable analysis to calculate p-values and odds ratios evaluating associations between 14 behavioral factors and seropositivity. Adjusted p-values were calculated using Bonferroni’s correction to account for multiple comparisons. The full reproducible code is available at https://github.com/t-gin/SARS-CoV-2_in_housepets.

Results

Animal demographic data and overview of sample types

Table 2 provides an overview of the number of each sample type (exposed PCR and serum samples, unexposed PCR controls, and pre-2019 serological controls). Sixty-four house pets from 32 households with SARS-CoV-2-PCR-positive humans were enrolled (exposed PCR and serum samples), and six house pets from households with SARS-CoV-2-PCR-negative humans were enrolled only in the PCR portion of the study as negative controls (unexposed PCR controls). The 43 pre-2019 serological controls came from 24 dogs and 19 cats. Figure 1 shows the types of samples collected from SARS-CoV-2-positive households by species.

Table 2 Overview of house pets sampled and samples analyzed for SARS-CoV-2 PCR and serology in dogs and cats from SARS-CoV-2-positive homes, SARS-CoV-2-negative homes, and pre-2019 banked serum.

	Exposed PCR and serum samples	Unexposed PCR controls	Pre-2019 serological controls	
Number of households	32	6	NA	
Number of dogs and cats	43 dogs
21 cats	3 dogs
3 cats	24 dogs
19 cats	
Breeds (n)	Dogs
• Mixed breed (30)
• Pug (3)
• Miniature poodle (2)
• Border collie (1)
• Pitbull (1)
• German shepherd dog (1)
• Pomeranian (1)
• Siberian husky (1)
• German shorthaired pointer (1)
• Newfoundland (1)
• Soft-coated wheaten terrier (1)
Cats
• Domestic shorthairs (19)
• Domestic longhairs (2)	Dogs
• Viszla (1)
• Whippet (1)
• Unknown (1)
Cats
• Domestic shorthair (2)
• Unknown (1)	NA	
Median age (age range)	All animals
• 6 years (3 months to 15 years)
Dogs
• 6 years (3 months to 15 years)
Cats
• 4 years (2 to 12 years)	Dogs (list of ages)
• 6 years
•10.5 years
• Unknown Cats (list of ages)
•6 months
•4.5 years
• Unknown	NA	
Number of swabs (Number oral, number rectal)	157 (104 oral, 53 rectal)	10 (6 oral, 4 rectal)	None	
Number of serum samples (Number of dogs, number of cats)	38 (24 dogs, 14 cats)	None	43 (24 dogs, 19 cats)	
Notes.

Abbreviation NA not applicable

Figure 1 Sample types collected by species

Breakdown of exposed PCR samples (oral/rectal) and serum samples collected for testing in dogs and cats from SARS-CoV-2-positive households in North Carolina.

Exposed PCR samples were collected from day 0 through day 60, with most swabs taken at day 0 (41 oral, 19 rectal) and day 28 (34 oral, 22 rectal). Thirty-five house pets (9 cats and 26 dogs) had oral or rectal swabs taken at more than one time point.

Thirty-nine exposed serum samples were submitted for antibody testing. One sample was removed due to suspected erroneous results, as indicated by an outlier during visualization (Fig. 2), resulting in 38 samples for analysis. This included exposed serum samples from 24 dogs and 14 cats collected on days 28 (n = 13 samples), 60 (n = 2 samples), 120 (n = 22 samples), and 180 (n = 2 samples).

Figure 2 Round 1 vs Round 2 optical densities for canine and feline anti-SARS-CoV-2- IgG and IgM

Anti-SARS-CoV-2 antibody optical density plots for canine IgG (A), canine IgM (B), feline IgG (C), and feline IgM (D). Serum samples were submitted for animals enrolled in the study (orange dots) at two separate time points and plotted on the x-axis (Round 1) and y-axis (Round 2). Samples that did not have enough serum for a second run were not plotted. Dog and cat pre-2019 control samples were submitted alongside the round 2 samples. Optical densities for the control samples (black dots) were set to equal on the x- and y-axis and plotted alongside the real samples.

Evaluating the ELISA to generate high confidence seropositivity calls

The anti-SARS-CoV-2 IgG and IgM O.D.s from replicate testing for the exposed serum samples were evaluated alongside the O.D.s for the pre-2019 serological controls. Four of the 38 exposed serum samples (one from dogs and three from cats) did not have adequate volume for two aliquots; these were excluded from the analysis to establish an O.D. cut-off. To visually assess for quantitative consistency, O.D.s for round 1 and round 2 serology from the exposed serum samples were plotted in Fig. 2, as well as O.D.s of the pre-2019 serological control samples. Additionally, Lin’s concordance correlation coefficient was calculated to measure agreement between the round 1 and round 2 O.D.s for dog and cat IgG and IgM. The correlation coefficient for dog IgG was 0.86, cat IgG was 0.96, dog IgM was 0.64, and cat IgM was 0.85, indicating relatively consistent measurements between both runs for dog and cat IgG and cat IgM, but not dog IgM. On initial visual assessment of the 34 round 1 and round 2 samples, two distinct clusters were identified for dog and cat IgG. A line was drawn as a proposed cutoff point to distinguish samples with O.D.s that were visibly higher or lower. The pre-2019 serological controls were then plotted over the exposed serum samples, and the previously drawn cutoff line was adjusted such that a large majority of the samples above the line were exposed serum samples. The chosen cutoffs represent one option for delineating between the distinct clusters of exposed serum sample O.D.s and minimizing the number of pre-2019 serological controls above the cut-off line (false positives). Dog and cat IgM samples did not cluster such that a distinct cutoff point could be established even before adding the control samples. Figure 2 graphically represents this, where positive samples cluster distant from negative pre-2019 serological controls and negative exposed house pet samples.

False positive rates

With the new cutoff points chosen, the false positive rate for IgG seropositivity in our pre-2019 serological controls was 0/24 (0%) in dogs and 1/19 (5.3%) in cats. A false positive rate for IgM was not calculated, as samples did not cluster such that a distinct cutoff point could be established. This contrasts with the ELISA cut-off procedure described earlier in our methods section, as well as in a previous manuscript, which resulted in a false positive rate for IgG seropositivity in the pre-2019 serological controls of 3/24 (12.5%) in dogs and 6/19 (31.6%) in cats. (Bienzle et al., 2022) Based on that same ELISA cut-off procedure, the false positive rate of IgM for the pre-2019 serological controls was 8/24 (33.3%) in dogs and 7/19 (36.8%) in cats.

Serology results

Based on the seropositive thresholds established with the pre-2019 serological controls, 9/24 (37.5%) dogs and 4/14 (28.6%) cats were IgG-positive. A breakdown of positive samples by timepoint includes 7/13 (53.8%) day 28 samples, 0/2 (0%) day 60 samples, 6/22 (27.3%) day 120 samples, and 1/2 (50%) day 180 samples. A summary of results from seropositive animals is provided in Table 3.

Table 3 Summary of SARS-CoV-2-IgG-positive dogs and cats from households with SARS-CoV-2- PCR-positive owners.

House number	Species	Timepoint (Days)	IgG result	
3	Feline	120	Positive	
7	Canine	120	Positive	
9	Canine	120	Positive	
10	Canine	120	Positive	
14	Canine	120	Positive	
15	Canine	180	Positive	
23	Canine	28	Positive	
25	Feline	28	Positive	
25	Feline	28	Positive	
25	Feline	28	Positive	
27	Canine	28	Positive	
28	Canine	28	Positive	
32	Canine	28	Positive	

PCR results

Three of 64 (4.8%) animals from 2/32 (6.3%) SARS-CoV-2-positive households were deemed SARS-CoV-2 positive based on a positive result from the QuantStudio 3 RT-PCR and the COBAS 6800 analyzer. Table 4 provides testing information on all five positive samples from the three animals mentioned, as well as one QuantStudio 3 RT-PCR-positive, COBAS-negative sample, which came from a cat in the same household as one of the QuantStudio 3 RT-PCR-positive dogs. The QuantStudio 3-PCR-positive, COBAS-negative cat had no target 1 or 2 genetic material detected on the COBAS analyzer. All nine of the IgG-positive dogs and all four of the IgG-positive were found to be RT-PCR-negative by the QuantStudio 3 between days 0–28. Among the three QuantStudio 3-RT-PCR-positive animals, two underwent serological testing, both on day 60. Neither animal was positive for anti-SARS-CoV-2 IgG.

Table 4 Summary of QuantStudio 3-RT-PCR-positive, COBAS-tested samples from dogs and cats in SARS-CoV-2-PCR-positive households.

Household	Species	Sample	Timepoint (day)	Viral load (RNA cp/mL)	QuantStudio 3 PCR result	Target 1 Ct value	Target 2 Ct value	COBAS result	
5	Feline	Oral	28	2399	Positive	–	–	Negative	
5	Canine	Rectal	28	1170	Positive	34.3	36.55	Positive	
22	Feline	Oral	0	3333	Positive	33.08	34.54	Positive	
22	Feline	Rectal	0	75	Positive	33.3	35.1	Positive	
22	Canine	Oral	0	592	Positive	34.5	35.75	Positive	
22	Canine	Rectal	w0	206	Positive	–	36.44	Positive	
Notes.

Abbreviations PCR polymerase chain reaction

RNA ribonucleic acid

Ct cycle threshold

Analysis of questionnaire data related to seropositivity

Of the 38 animals with serology results, 36 (23 dogs, 13 cats) had a complete Questionnaire 2, and 38 (24 dogs, 14 cats) had a complete Questionnaire 3 from the time of sample collection. Questionnaires were analyzed with serum data but not PCR data due to the large number of missing questionnaires from animals with PCR but not serology and very few PCR-positive animals.

Fourteen exposures were tested for a relationship with IgG seropositivity in dogs and cats. Associations and calculated statistics are reported in Table 5. No statistically significant associations were found in cats. In dogs, prior to applying Bonferroni’s correction, a statistically significant positive association was found between an IgG-positive ELISA and the owner reporting that dogs were allowed on the owner’s bed (p = 0.04) or furniture (p = 0.048). A negative association was identified between dogs within IgG-positive ELISA and owners reporting that dogs were known to lick plates in the dishwasher (p = 0.04). None of these associations were found to be significant following Bonferroni’s correction, which is represented by the “adjusted p-value” in Table 5.

Table 5 Associations measured between SARS-CoV-2-seropositive and seronegative dogs and exposure of interest based on owner questionnaire.

Exposure	n/N (%)
IgG positive	n/N (%)
IgG negative	Odds ratio	p-value	Adjusted p-value	
Contact with owner’s bed	8/9 (88.9%)	6/14 (42.9%)	9.61	0.040*	0.56	
Known to lick plates in dishwasher	1/9 (11.1%)	8/14 (57.1%)	0.10	0.040*	0.56	
Contact with furniture	9/9 (100%)	8/14 (57.1%)	INF	0.048*	0.67	
Known to lick owner’s face	5/9 (55.6%)	7/14 (50%)	1.24	1.00	1.00	
Known to lick owner’s hands	6/9 (66.7%)	12/14 (85.7%)	0.35	0.34	1.00	
Known to lick owner other than face or hands	8/9 (88.9%)	8/14 (57.1%)	5.6	0.18	1.00	
Eats off owner’s plate	6/9 (66.7%)	7/14 (50%)	1.94	0.67	1.00	
Hunts or brings prey	1/9 (11.1%)	1/14 (7.1%)	1.59	1.00	1.00	
Bitten owner within last month	2/9 (22.2%)	0/14 (0%)	INF	0.14	1.00	
Scratched owner within last month	5/9 (55.6%)	5/14 (35.7%)	2.17	0.42	1.00	
Classification of home area suburban (not urban)	7/9 (77.8%)	14/14 (100%)	0	0.14	1.00	
Known to eliminate in-home	3/9 (33.3%)	5/14 (35.7%)	0.90	1.00	1.00	
Sex of animal is female	2/9 (22.2%)	8/15 (53.3%)	0.27	0.21	1.00	
Animal neutered	8/9 (88.9%)	14/15 (93.3%)	0.59	1.00	1.00	
Notes.

Abbreviations IgG immunoglobulin G

Conclusions regarding symptom data were limited by the statistically underpowered number of observations of each symptom in the dataset. As such, symptom data was not found to be significantly associated with a positive IgG serology (p = 0.40). Three out of nine (33.3%) IgG-positive dogs were reported to have symptoms at one or more timepoints during sample collection. A table of the testing timeline and results for each of these dogs is included in supplementary materials (Supplemental Information 5). No symptoms were reported in any house pet with a positive SARS-CoV-2 PCR. No symptoms were reported in any cats with serology performed.

Discussion

Our study detected an IgG seroprevalence of 37.5% and 28.6% in dogs and cats, respectively, in the Raleigh-Durham area living with a SARS-CoV-2-infected person/people when sampled 28-180 days after a positive test in a human household member. The number of SARS-CoV-2 PCR-positive samples from animals in homes with SARS-CoV-2-PCR-positive individuals was low (3/64, 4.7%) but within the range of PCR-positive animals from studies of a similar design. No stastitically significant associations were identified between questionnaires related to owner-reported animal symptomology or behavior and IgG-seropositivity after correcting for multiple hypotheses.

A key finding in this study is that in the SARS-CoV-2 ELISA performed here, IgG in dogs had the clearest and most consistent results compared to IgM in dogs, and IgG and IgM in cats, based on distinct clustering of results from exposed serum samples between two runs and when compared to pre-2019 serological controls (Fig. 2). There was no distinct difference between IgM results in pre-2019 serological controls and exposed serum samples (cats or dogs). This may be because SARS-CoV-2 IgM antibodies were not present at above baseline levels in these pets, as IgM is typically present earlier in the course of seroconversion, or because the assay lacks sensitivity for true differences. Additionally, IgM is notoriously nonspecific. This brings into question whether IgM should be reported for either species in future studies unless extensive validation, for instance, the use of multiple population-representative negative controls, is used to support the results. Additionally, based on our results, IgG ELISA for SARS-CoV-2 in cats should be interpreted particularly cautiously and only after validation with negative and positive controls has resulted in known sensitivity and specificity for that particular test. When attempting to determine if a dog has been exposed to SARS-CoV-2, IgG ELISA appears preferable to IgM ELISA at the 28-day or later timepoint. In any case, where novel therapeutics are used to detect disease exposure, it is essential to understand how the test was developed and the types of cutoffs used, and ideally for population-representative negative controls to be used to provide an understanding of population-level variation. Between runs, we also identified technical variation most notably in dog IgM, but also in dog and cat IgG and cat IgM. Despite the technical variation identified (and quantified), we were able to identify a clear distinction between positive and negative animals through the use of negative control samples. Knowledgeof such test parameters is essential for interpreting serology results reported in the literature, especially if it is not specified whether IgG or IgM was detected. When trying to determine an optimal cut-off point based on the pre-2019 serological controls, we attempted to minimize the number of false positive control samples while honoring the clear distinction between groups of suspected positive and negative exposed serum samples. No single cutoff would be perfect in this case, as choosing a higher cutoff would almost certainly result in false negative results, and a lower cutoff would increase false positive results.

Interestingly, two of our PCR-positive animals were IgG-negative at day 60. However, this may represent contamination of SARS-CoV-2 RNA from the household family members or environment rather than true infection. Regarding the PCR-positive animals, a Ct < 37 for targets 1 and 2 was used as the cutoff for determining a positive COBAS 6800 result. Newer research since the time of our sample testing suggests that a Ct < 33 for both targets is more specific for SARS-CoV-2 (Grewal, Syed Gurcoo & Sudhan Sharma, 2022). The fact that none of our samples were <33 (Table 4), and at least one sample was only positive for target 2, further adds to our theory that environmental contamination is more likely than true infection in our population of animals.

Compared to other North American studies of a similar design, our IgG seroprevalence falls on the upper range for dogs (11–41%) and the lower end of the range for cats (21–51%) (Newman et al., 2020; Barua et al., 2021; Dileepan et al., 2021; Hamer et al., 2021; Murphy & Ly, 2021; Fritz et al., 2021; Cossaboom et al., 2021; Goryoka et al., 2021; Bienzle et al., 2022). The variability in seropositivity between studies is likely multifactorial, involving differences in sampling methods, testing methods, geographic distribution, varying analyses of positive and negative samples, or potentially study timepoint in relation to COVID-19 surges. Our study sampled blood for serology between days 28 and 180 post-owner diagnosis, whereas previous North American studies have reported (when documented) serum sample collection 3–42 days post-exposure. The timeline for sampling in our study may have allowed for a longer period of time for seroconversion following exposure to SARS-CoV-2, and, of clear importance as shown here, the measurement methodology varied.

We attempted to ascertain relationships between seropositive pets and characteristics of the pet-owner relationship through questionnaires. Prior to correcting for multiple comparisons, we found significant positive relationships between dogs allowed on the bed or furniture and IgG-seropositivity, as well as a significant negative relationship between dogs that lick plates in the dishwasher and IgG-seropositivity. None of these behaviors were found to be significant after Bonferroni’s correction. However, the first two associations may have merit in terms of a relationship, as it would make sense that dogs that share furniture and bedding are more likely to come into contact with the virus. With that in mind, the positive association between dogs being allowed on the couch or bed and having a higher likelihood of seropositivity aligns with findings in a previous study and thus may warrant further exploration.(Bienzle et al., 2022). We did not find a statistically significant association between owners sharing food with their pets and seropositivity as has been shown previously (Alberto-Orlando et al., 2022). Similarly, the idea that dogs that lick dishes in the dishwasher had a much lower rate of IgG-seropositivity is most likely indicative of type 1 error, as there is no plausible biologically reasonable explanation for why this would occur. Future studies may benefit from further exploring these behavioral associations with a larger sample size.

Our study was strengthened through the use of negative controls and repeat testing of the same samples in the same lab using the same methodology. However, our study is not without limitations. The small sample size may limited our power to identify relationships between behavioral factors and seropositivity. This is highlighted by the fact that, prior to Bonferroni’s correction, licking plates in the dishwasher had a negative relationship with SARS-CoV-2 seropositivity in dogs, and contact with furniture had a positive relationship with seropositivity. Although not demonstrated in this study, it is possible that such relationships could be validated through larger sample sizes.

In addition to sample size, our study would have benefited from using multiple different types of control samples, including pre-2019 negative controls for PCR, known positive serum samples for calculating a false negative rate, and swab and serum samples positive for other types of coronaviruses to assess for cross-reactivity. Unfortunately, we were unable to locate such sample types and thus attempted to increase the strength of determining whether an animal was positive for viral genes through the use of two testing modalities (RNA and the COBAS 6800 system). Moreover, we did not collect data on whether dogs and cats had been previously vaccinated for other coronaviruses which theoretically could cross-react with SARS-CoV-2 serology. Additionally, we were limited by the availability of owners and willingness to participate at any given time, which resulted in a variable sampling timeline. Lastly, the results of this study reflect early patterns of the SARS-CoV-2 (prior to any variants) and may not represent the role of companion animals with the current SARS-CoV-2 variants. Nevertheless, the results emphasize the importance of using robust control methods when validating and reporting test results.

This study did not generate strong evidence to support or negate animals as a significant reservoir for transmission of SARS-CoV-2, although that was not a specific aim. However, we brought into question the validity of “positive” and “negative” results reported for SARS-CoV-2 ELISAs that have not been evaluated with a large number of negative controls, specifically IgM in dogs and cats and IgG in cats. This is demonstrated with the false positive rates calculated based on the cutoff points we established versus the much higher false positive rates reported from the initial ELISA procedure.

With respect to serology, virus neutralization (V.N.) could have contributed to the strength of our findings, as it is often considered a gold standard for specific antibody activity. Still, the lack of V.N. does not necessarily take away from the main finding that SARS-CoV-2 ELISA should be thoroughly investigated prior to reporting. Virus neutralization could be considered for parallel sampling of pets when investigating shared infections in the future.

Conclusions

The main goal of our study was to evaluate SARS-CoV-2 PCR and serology in a population of dogs and cats living in households with SARS-CoV-2-PCR-positive humans, as well as provide robust analytical methods for validating these results. Few (3/64) animals were PCR-positive for SARS-CoV-2 RNA, despite living in homes with known SARS-CoV-2-PCR-positive humans, and had high Ct values and no documentation of seroreactivity, supporting the possibility that these PCR positives were all compatible with environmental contamination. A larger number of animals (13/38), including 38% of dogs (9/24) and 29% of cats (4/14) were seropositive for IgG, as determined through robust ELISA validation methods. At the day 28 sampling timepoint, over 50% of sampled pets were IgG seropositive (7/13).

Supplemental Information

Supplemental Information 1 Information about PCR methodology used for this study

Click here for additional data file.

Supplemental Information 2 Questionnaire 1: reviews enrolled animal demographic information

Click here for additional data file.

Supplemental Information 3 Questionnaire 2: inquires about symptoms present in canine and feline participants

Click here for additional data file.

Supplemental Information 4 Questionnaire 3: reviews questions about the human-animal interactions within each household

Click here for additional data file.

Supplemental Information 5 Overview of testing timeline and symptoms present in animals with positive IgG serology for SARS-CoV-2 and the presence of owner-reported symptoms on at least one timepoint

Click here for additional data file.

Supplemental Information 6 Raw sample data from SARS-CoV-2-tested dogs and cats

Click here for additional data file.

Supplemental Information 7 Raw patient data from dogs and cats in SARS-CoV-2-positive households tested for SARS-CoV-2

Click here for additional data file.

The authors would like to thank Edward B. Breitshcwerdt, Maria Nadworny, Christina Nix, Jack Anderson, T. Scott Alderman, Thad Gurley, Raul Louzao, Rosemarie Asrican.

Additional Information and Declarations

Competing Interests

Author Contributions

Human Ethics

Animal Ethics

Field Study Permissions

Data Availability

Dorothee Bienzle is an Academic Editor for PeerJ.

Taylor E. Gin conceived and designed the experiments, performed the experiments, analyzed the data, prepared figures and/or tables, authored or reviewed drafts of the article, and approved the final draft.

Elizabeth A. Petzold conceived and designed the experiments, performed the experiments, authored or reviewed drafts of the article, and approved the final draft.

Diya M. Uthappa analyzed the data, authored or reviewed drafts of the article, and approved the final draft.

Coralei E. Neighbors analyzed the data, authored or reviewed drafts of the article, and approved the final draft.

Anna R. Borough analyzed the data, authored or reviewed drafts of the article, and approved the final draft.

Craig Gin analyzed the data, authored or reviewed drafts of the article, and approved the final draft.

Erin Lashnits conceived and designed the experiments, authored or reviewed drafts of the article, and approved the final draft.

Gregory D. Sempowski performed the experiments, authored or reviewed drafts of the article, and approved the final draft.

Thomas Denny performed the experiments, authored or reviewed drafts of the article, and approved the final draft.

Dorothee Bienzle performed the experiments, analyzed the data, authored or reviewed drafts of the article, and approved the final draft.

J. Scott Weese performed the experiments, authored or reviewed drafts of the article, and approved the final draft.

Benjamin J. Callahan analyzed the data, authored or reviewed drafts of the article, and approved the final draft.

Christopher W. Woods conceived and designed the experiments, authored or reviewed drafts of the article, and approved the final draft.

The following information was supplied relating to ethical approvals (i.e., approving body and any reference numbers):

Duke Institutional Review Board granted ethical approval to carry out this study.

The following information was supplied relating to ethical approvals (i.e., approving body and any reference numbers):

Duke Institutional Animal Care and Use Committee provided full approval for this research.

The following information was supplied relating to field study approvals (i.e., approving body and any reference numbers):

Samples were collected in the homes of owners, which was approved by the Duke IRB and IACUC.

The following information was supplied regarding data availability:

The code is available at GitHub: https://github.com/t-gin/SARS-CoV-2_in_housepets.

The data are available at Mendeley: Gin, Taylor (2023), “Evaluation of SARS-CoV-2 identification methods through surveillance of companion animals in SARS-CoV-2-positive homes in North Carolina, March to December 2020”, Mendeley Data, V1, doi: 10.17632/czvdjmscj8.1.

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
