# Peer review of "Evaluation of SARS-CoV-2 identification methods through surveillance of companion animals in SARS-CoV-2-positive homes in North Carolina, March to December 2020"

_PeerJ, doi:10.7717/peerj.16310_

## Round 0.1 · original submission · Minor Revisions

This is a well-written manuscript with only a few minor issues that need to be addressed. The authors did a great job of explaining the study and results. There are just a few minor issues with flow and some additional data requested to be added to the abstract.

Reviewer 1 ·

Basic reporting

Clear and concise; sufficient background information provided; literature cited properly; figures/tables self explanatory and complete; conclusions justified.

Experimental design

Standard methods used; question well-defined; important topic of investigation; enough information provided for repeating experiments

Validity of the findings

Results are clearly presented; conclusions are justified

Additional comments

The manuscript by Gin et al reports on SARS-CoV-2 surveillance of companion animals who lived in households with infected humans. This study utilized serological and molecular assays for the investigation. A 4.8% (3/64) were positive by RT-PCR and 28.6% and 37.5% of cats and dogs were positive for the pathogen specific IgG antibodies. The study is important in understanding various aspects of SARS-CoV-2 infection in pets.

Here are some comments/questions:

1. Is there any information on potential cross-reactivity of the antigen used in ELISA in this study with other coronaviruses?

2. Can the cut-offs generated using pre-2019 population controls still be used, especially considering the fact that a large number of animals and humans have since been exposed to the virus and that might have changed the baseline values in the population?

3. What single sample in pets (nasal vs. oral vs. rectal) do the authors consider as the ‘best’ for molecular diagnosis of SARS-CoV-2?

4. In the last sentence of abstract...."A small number of pet dogs and cats...." I suggest that a percent positivity (xy%) should be provided for the RT-PCR results as well.

·

Basic reporting

no comment

Experimental design

limited sample sized

Validity of the findings

The title
reflects a greater emphasis on sars cov 2 surveillance in animals in homes with positive owners, but note that it is a small group of homes with greater emphasis on identification methods.

Introduction
I think it is important to present a brief sketch of the situation (time and space) of sars cov 2 in humans during the sampling phase in the study locality.
In the questionnaire we asked questions relevant to the mechanism of acquisition of the virus as performed by Solon Alberto-Orlando, et al; SARS-CoV-2 transmission from infected owner to household dogs and cats is associated with food sharing, International Journal of Infectious Diseases, Volume 122, 2022, Pages 295-299, ISSN 1201 9712, https://doi.org/10.1016/j.ijid.2022 .05.049. (https://www.sciencedirect.com/science/article/pii/S1201971222003137)
I think that they could be mentioned for later discussion and constructive criticism of the papers presented.

Additional comments

it is a great job

---

## Round 0.2 · Minor Revisions

The authors adequately address the comments from the reviewers and editor. However, there are a few minor editorial changes that need to be addressed. The citation format does not follow the PeerJ guidelines. The citation should be before the period not after. Additionally, numbers 1-9 should be written out unless they are associated with units.

---

## Round 0.3 · accepted · Accept

Good job addressing the editorial issues.